# Circulating Tumour Cells in the Prediction of Bone Metastasis

**DOI:** 10.3390/cancers16020252

**Published:** 2024-01-05

**Authors:** Siu-Wai Choi, Aria Kaiyuan Sun, Jason Pui-Yin Cheung, Jemmi Ching-Ying Ho

**Affiliations:** 1Department of Orthopaedics and Tramatology, School of Clinical Medicine, Li Ka Shing Faculty of Medicine, The University of Hong Kong, Hong Kong SAR, China; 2Department of Anaesthesiology, School of Clinical Medicine, Faculty of Medicine, The University of Hong Kong, Hong Kong SAR, China; u3008500@connect.hku.hk (A.K.S.); jemmiho@hku.hk (J.C.-Y.H.)

**Keywords:** circulating tumour cells, CTCs, bone metastasis, BM, cancer prediction

## Abstract

**Simple Summary:**

Although primary bone cancer makes up only 1% of worldwide cancer cases diagnosed annually, the bone is the most common site of metastasis for tumours of the breast, prostate and lung. Once a tumour has metastasised to the bone, treatment is predominantly palliative. Though metastasis is the main cause of mortality in cancer patients, the mechanisms of metastases are not well understood. It is known that cells released from the primary tumour into the circulation (Circulating Tumour Cells, CTCs) may play a role in causing disease spread. This review summarises the technologies used to isolate and count CTCs, and the studies investigating the role of CTC numbers and CTC cell surface protein expression in the prediction of cancer metastasis to the bone.

**Abstract:**

Bone is the most common organ for the development of metastases in many primary tumours, including those of the breast, prostate and lung. In most cases, bone metastasis is incurable, and treatment is predominantly palliative. Much research has focused on the role of Circulating Tumour Cells (CTCs) in the mechanism of metastasis to the bone, and methods have been developed to isolate and count CTCs from peripheral blood. Several methods are currently being used in the study of CTCs, but only one, the CellSearchTM system has been approved by the United States Food and Drug Administration for clinical use. This review summarises the advantages and disadvantages, and outlines which clinical studies have used these methods. Studies have found that CTC numbers are predictive of bone metastasis in breast, prostate and lung cancer. Further work is required to incorporate information on CTCs into current staging systems to guide treatment in the prevention of tumour progression into bone.

## 1. Background

Cancer is a disease caused by changes in cellular DNA leading to the uncontrolled division of cells. According to the World Cancer Research Fund, there were 18.1 million cancer cases worldwide in 2020 alone [1]. Although bone cancer is relatively rare, accounting for only 1% of all cancer cases [2], it is the most frequent site of cancer metastasis for many of the most commonly occurring cancers; up to 65–75% of patients with breast and prostate cancers develop bone metastases during the course of their disease, and about 15–40% of lung, thyroid, stomach and kidney cancers develop bone metastases [3,4,5,6,7,8,9,10,11,12].

Metastasis is when a secondary tumour develops at a site that is anatomically distinct from the site of the original tumour. Although metastasis is the main cause of mortality in cancer patients, the mechanisms of metastases are poorly understood. It is known that large numbers of cells can dissociate from the primary tumour and enter the circulation, carrying malignant cells and seeding tumours at distant sites. Despite the fact that large numbers of cells are dissociated from the primary tumour daily, only a very small number (<0.1%) do actually result in a new tumour growth [13,14]. This is not surprising, considering the tribulations experienced by a dissociated tumour cell upon entering the blood circulation, which includes high pressure [15], shear stress [16] and immune surveillance [17].

Although such a small proportion of these circulating tumour cells (CTCs) which have entered the circulation can survive and seed a metastasis, distant metastatic disease accounts for over 90% of cancer deaths or treatment failure in many malignancies [18]. While the exact mechanisms of how tumours metastasize is still unknown, it is known that the bone is a particularly common metastatic site [19]. Figure 1A–C summarise how CTCs enter the circulation and cause a metastasis in the bone. It is known that up to 40% of lung and urinary system tumours will metastasise to the bone, and this figure is even higher for cancers of the breast, with over 70% metastasising to the bone [20,21,22]. One-year survival for these patients is dismal, with only 10% of lung and 50% of breast-to-bone metastasis patients surviving more than a year [3]. In order to improve survival in those patients suffering from malignancies possessing a specific propensity to migrate to the bone, it is incumbent to understand, and prevent, bone metastasis.

Despite the high incidence of bone metastasis in common cancers, it is still not well understood why dissociated tumour cells are particularly osteotropic. Studies to optimise bone marrow transplantation have found that the bone vasculature differs from that of other organs. In particular, the vasculature in the metaphysis of the long bones constitutively express adhesion molecules including P-selectin, E-selectin, as well as intercellular and vascular adhesion molecules [23,24], whereas vasculature of other organs only express such molecules when in an inflammatory state [25]. The expression of these adhesion molecules facilitates interaction of the blood vessels with circulating tumour cells, and because the metaphysis vasculature is fenestrated [26], this may facilitate the intravasation of the CTCs into bone marrow stroma. Circulating tumour cells (from breast tumours), on their part have been shown to secrete angiogenic factors and microRNA-105, both of which can decrease the expression of tight junction proteins of the vascular endothelial layer [27]. This decrease in tight junction proteins will enable easier extravasation of CTCs into an organ, and along with the bone marrow vascular fenestration, CTCs will have no difficulty leaving the circulation and embedding into the bone.

Bone marrow vasculature has another unique property—the expression of stromal cell-derived growth factor-1 (SDF-1) in specific areas. The expression of SDF-1 is an important factor in the osteotropic tendency of CTCs since CTCs express high levels of CXC chemokine receptor 4 (CXCR4), in breast cancer cells at least, which is the ligand for SDF-1 [28]. These findings show that CTCs have a specific homing tendency to the bone vasculature and the microenvironment of the bone.

Clinically, the current diagnostic methods for bone metastasis include bone scans, whole body magnetic resonance imaging (MRI) and positron emission tomography (PET) [29]. However, these technologies are very expensive, only available in larger institutions and can only detect obvious metastases. These limitations may result in a delay in the diagnosis of bone metastasis, and ultimately postpone treatment.

An early indicator of metastatic potential is to enumerate CTCs from patient blood samples. The advantages of using CTCs over traditional imaging methods are that it is safe, does not require administration of drugs (such as contrast), non-invasive and can be conducted repeatedly over the course of treatment. In this regard, studies investigating cancer stage, cancer progression, metastases and response to treatment in relation to CTC numbers have been conducted. This current review therefore aims to provide a comprehensive analysis of clinically relevant studies and the latest CTC detection technology in relation to bone metastasis.

### 1.1. Search Strategy and Eligibility Criteria

This systematic review provides a comprehensive overview of the latest CTC detection technologies and all studies relating to the detection of bone metastases with regard to the action of CTCs. Searches were conducted using PubMed and the Web of Science database, with keywords set to “CTC” OR “CTCs” OR “Circulating tumour cells” AND “Bone metastasis” OR “BM” OR “Bone cancer”, published in English without limit on year of publication. Inclusion criteria are original articles of cohort studies reporting the detection of CTCs in patients diagnosed with bone metastases, and only articles at the highest or second highest level of evidence for prognostic studies which are ‘high quality prospective or retrospective studies with adequate power’ have been reviewed here. Studies reporting the use of other circulating tumour biomarkers, such as ctDNA, and studies in animals, were excluded from this review. Publications which were considered lower levels of evidence such as expert opinions and commentaries have also been excluded.

### 1.2. Latest Screening Techniques for CTCs in Patients Diagnosed with Bone Metastasis

Circulating tumour cells in the blood are rare, and the numbers are variable in comparison to the hematopoietic cells. Studies have found approximately 10 CTCs per 10 mL of blood in typical cancer patients, compared to over sixty million blood cells in the same volume [5]. Thus, validated enrichment methods, which aim to reduce the sampled blood volume and concentrate the CTCs for detection, are required before these cells can be enumerated. Current CTC detection technologies are usually categorized based on biophysical-based technologies such as tapered slit filter platforms, or immunoaffinity-based technologies such as CellSearch™ systems and polymerase chain reactions [30]. Studies using blood sampled from patients with bone metastasis are summarized in Table 1.

Currently, the CellSearch™ System is the only U.S. Food and Drug Administration approved method for the detection of CTCs in biological samples. The CellSearch™ system relies upon immunoaffinity to isolate CTCs which uses antibody binding to cell surface markers on CTCs. The two main antibodies used are epithelial cell adhesion molecules (EpCAM) and cytokeratins [30]. The CellSearch™ system requires only a small amount of blood (7.5 mL) and therefore limits the clinical burden on patients. Using the CellSearch™ system (Cell Search, Huntington Valley, PA, USA), CTC counts have been used for the diagnosis and prognosis of bone metastasis in breast, lung, prostate and neuroendocrine cancer [30,31,32,33] and a significant association between the presence of CTCs and bone metastasis has been found in lung and neuroendocrine cancer [32,33]. There are, however, limitations of the CellSearch™ system, which include the high equipment costs and the CTCs themselves being highly variable in terms of cell surface markers expressed. Some CTCs may not even express EpCAM and therefore would not be identified using the Cellsearch™ system.

The CTC iChip technology (using 10 mL of blood) is similar to the CellSearch™ system in that it also utilizes Immunomagnetic enrichment to distinguish CTCs from other cells in the blood. Using the CTC iChip technology, CTCs could be detected in 59% of breast cancer patients who had been diagnosed with new bone metastases. In patients with progressing bone metastasis, an increased expression of cytokines from CTCs and epithelial-mesenchymal transition regulators for example, androgen signalling, were detected. The preparation of the CTC iChip kit is expensive and labour intensive as different segments must be assembled separately, this makes using the CTC iChip time consuming and impractical for large numbers of patient samples [38].

Unlike the Cellsearch™ system and CTC-iChip technologies, RT-PCR utilizes specific CTC transcript (mRNA) characteristics to isolate CTCs [37]. In castration-refractory prostate cancer, CTCs expressing kallikrein-related peptidase 3 mRNA were only detected in patients with a clinical diagnosis (through soft-tissue imaging and bone scans) of bone metastasis, which indicates that this peptidase can be used as an early bone metastatic detection [37]. A disadvantage of using RT-PCR is that the isolated CTCs have to be lysed in order to extract RNA and therefore cellular morphology cannot be ascertained.

Although the set-up for isolation of CTCs is expensive and labour intensive, studies comparing the cost of imaging and CTC isolation have found that the detection of CTCs is still a relatively inexpensive tumour marker which can be incorporated into routine cancer screening [39]. Though currently, the Cellsearch™ system is the only FDA-approved technology for CTCs detection, new and better technologies are constantly being developed for cheaper, less-labour intensive and more comprehensive isolation of CTCs.

### 1.3. Summary of Clinical Research Relating to CTCs Detection

A total of 24 articles fulfilled the inclusion criteria and have been included in this review from a total of 1762 articles searched (Table 2).

Existing research relating to CTCs and bone metastasis can be broadly divided into three foci, the analysis of differences in the application of different CTC capture modalities in order to develop methods suitable for clinical use, to explore the mechanisms through which CTCs cause metastasis to bone, and to predict patient prognosis based on CTC numbers.

### 1.4. CTCs as Prognostic Biomarkers of Recurrence and Therapeutic Targets for Bone Metastasis

The presence of CTCs has been shown to have a prognostic role in many cancer types, and though not all studies have shown statistical significance, putative evidence does indeed suggest that CTC numbers can be used as an early predictor of bone metastasis. A study of metastatic kidney cancer found that 39% of patients had bone metastases. Although CTC numbers were not able to differentiate those patients with metastasis to the bone, however, the investigators did find that those patients with bone metastasis were more likely to have more than three CTCs in their blood sample [31]. This finding was also corroborated in a study of 195 patients with recurrent/progressive metastatic breast cancer. CTC numbers were significantly higher in patients with bone metastases than in patients without bone metastases. It was also found that when bone metastases were present, 35% of patients had more than 21 CTCs and 61% had more than 5 CTCs; in patients without bone metastases, only 16% of patients had CTC counts greater than 5, showing a strong association between high CTC numbers and extensive skeletal involvement [30,32].

#### 1.4.1. Use of CTCs in Anti-Tumour Drug Screening

Other uses of CTCs include their use in anti-tumour drug screening. Cultured CTCs isolated from patients can be used to screen the effectiveness of a panel of chemotherapeutic drugs before they are applied to the patient. In vitro derived, fluorescently labelled CTCs can also be injected intravenously into animal models to track cells and to learn more about where CTC seeding and metastasis may occur, providing unequivocal evidence of the pivotal role CTCs play in metastasis [31,40].

#### 1.4.2. Gene Expression in CTCs

In addition to CTC enumeration and study of epitope expression, there has been continued interest in genes expressed by CTCs from primary tumours that preferentially replicate in bone. In this respect, studies have confirmed that the trilobin-encoding genes TFF1 and TFF3 are two of the highest ranked, most likely candidates’ genes, the expression of which may lead to cancer metastasis to bone [41,42]. In addition to the trilobin proteins, the expression of RANKL may also be a potential therapeutic target. Studies investigating the binding of RANKL to its receptor RANK and its role in the mechanisms of bone metastasis, have shown that the persistence of RANK-positive CTCs actually predicts a more favourable skeletal outcome. Area under the curve in survival analyses has shown that the persistence of RANK-positive CTCs (positive ΔAUC) during denosumab administration was associated with a delay in time to bone metastasis and metastatic progression, but not in delayed visceral metastasis progression [43].

In summary, not all studies have been able to find a relationship between the presence of CTCs and the prediction of bone metastasis. This could be due to the site origin of the primary tumour, as a variety of sites, including breast, prostate, lung and neuroendocrine tumours have been studied. However, in studies investigating primary breast cancer, it has been shown that a CTC count of ≥5 is predictive of a bone metastasis in 90% of patients. Even more interesting is that the CTC count is able to predict time to progression of bone metastasis in breast cancer patients from time of first diagnosis of the primary tumour. This finding in breast cancer has been reiterated in patients with primary tumour of the lung, where it has been found that a CTC count of ≥1 is predictive of a bone metastasis in 94% of patients.

The CTC predictive cutoff values in prostate primary tumours are less clear. In the study of prostate cancer patients, all patients with bone involvement had very high CTC counts (≥16), and it is not clear where a cutoff should be. This finding may be due to the fact that all patients were suffering from castration-resistant prostate cancer, which may be late-stage. Future studies are required to ascertain if CTCs can be predictive of bone metastasis in patients with early-stage prostate primary tumours.

Findings in neuroendocrine tumours are even more tantalizing and deserving of further research. Investigators have found that it is not the CTC count which is most important in bone metastasis prediction; rather, it is the expression of a specific osteotropic protein, CXCR4, which predisposes these CTCs to embed into bone and cause tumour cell growth.

#### 1.4.3. Changes in CTC Numbers Post Therapy

It is also important to note that interventional factors may also impact the potential prediction of bone metastasis using CTCs, these include surgery, chemotherapies, and immunotherapies. Firstly, although surgery is aimed at eradicating cancer, there is some evidence that it is paradoxical as it may cause an increase in CTCs. In one study, investigators comparing CTC levels in pre- and postoperative peripheral blood from patients with breast cancer found that the CTC positive rate increased after surgery, implying postoperative micrometastatic risk. The increase was more profound in the open surgery group (29.82% of patients) compared to the endoscopic group (15.09% of patients) [55]. On the contrary, investigators evaluating the CTC levels in blood samples taken from patients with non-small cell lung cancer saw a decrease in CTCs in the radial artery but not the pulmonary vein post-operation, suggesting central clearance. One study even suggests that CTC counts were independent of the surgical approach [56].

It is also important to note that CTC numbers may not decrease in a linear fashion following chemotherapy. In blood samples obtained from 1132 patients with breast cancer receiving adjuvant chemotherapy, significantly more patients converted from CTC-positive to CTC-negative than vice versa across different subtypes [57]. Furthermore, Ortiz-Otero et al. noticed a twofold increase in CTC numbers in 87% of patients with metastatic diseases following their first round of chemotherapy. However, after the second cycle of treatment, CTC levels eventually returned to baseline in 63% of patients. Therefore, there is also a possibility that chemotherapy evokes CTC mobilisation as part of the body’s systemic response.

Patients with non-small cell lung cancer who did not show any detected CTC or had decreasing CTC numbers following immunotherapy correlated with a more durable response to treatment, and the lack of disease progression at six months [58]. When analysing blood samples from these patients who were treated with checkpoint inhibitors, researchers noted that CTC presence was a predictive factor for a worse durable response rate [58]. In order to facilitate metastatic progression, CTCs must evade or suppress immune surveillance, employing strategies such as the formation of CTC-neutrophil clusters or the expression of PD-L1 [59]. This subsequently misleads the immune system, and T-cell activation is then inhibited.

#### 1.4.4. Use of CTCs in Combination with Other Biomarkers to Predict Bone Metastasis

Other biomarkers are associated with risk of bone metastasis in addition to CTCs, with the majority of these markers signifying the process of bone formation and bone resorption. In a meta-analysis investigating the levels of serum alkaline phosphatase (ALP) and bone-specific alkaline phosphatase (BALP) in patients with breast cancer, high levels of both markers were detected in the bone-metastatic group, confirming their potentials as early serologic markers. ALP is a group of isoenzymes expressed ubiquitously on the cell membrane and a high level is common amongst patients with metastasis [60]. As one of its subtypes, BALP is produced by osteoblasts and is a more accurate indicator of patients having larger volume or aggressive bony metastatic diseases. In blood samples obtained from patients, investigators noted that those with metastasis presented with significantly higher levels of BALP, which correlated with the number of bone metastatic sites [61]. In addition, to tightly regulate the concentration of calcium in the circulation, almost all calcium in the body exists as deposits in the skeleton. However, patients with bone metastasis may present with hypercalcemia. In patients with bladder cancer, researchers confirmed that besides elevated alkaline phosphatase levels, increased metastatic burden also correlated with heightened calcium, which is thought to play a role in mediating osteoblasts and osteoclasts differentiation [62].

N-terminal telopeptide (NTx) released during bone breakdown is a good marker of bone resorption and can be detected in serum and urine. Previous studies have offered differing conclusions on its effectiveness as a diagnostic tool for bone metastasis. A recent meta-analysis incorporating 62 eligible studies concluded that NTx is a useful biomarker for early diagnosis and prognosis of bone metastases for lung cancer, breast cancer and prostate cancer [63]. Tartrate-resistant acid phosphatase 5b (TRAP) is an enzyme that is solely produced by osteoclasts, with its levels proportional to the intensity of bone resorption. Researchers found that patients with breast and prostate cancers with skeletal involvement possessed significantly higher levels of serum TRAP5b, compared to patients without metastasis and healthy subjects [64].

Since its entry into routine use, prostate-specific antigen (PSA) has been considered an ideal indicator of bone metastasis in patients with prostate cancer, with a serum PSA value of >100 ng/mL signifying a greater likelihood of existing bone metastases at first diagnosis [65]. However, in a subsequent nationwide population-based study, it was not only concluded that the best PSA cut-off for predicting metastasis should be five-fold of its current guidelines, but even with this new cut off, the use of PSA should not replace the use of imaging [66]. Other less commonly used biomarkers include carcinoembryonic antigen (CEA), a fetal glycoprotein which is usually not produced in significant quantities after birth—an elevated concentration of this protein is hence a good indicator of malignancy development. At a cellular level, investigators have observed a correlation between serum CEA levels with colorectal metastasis [67]. The study confirmed that high serum CEA concentration was a potential risk factor for detecting bone metastasis, with an accuracy of 79.1%. Indeed, when combined with the use of ALP and CA125 levels, the specificity and accuracy for diagnosing bone metastasis from colorectal cancer reached 76.6% and 87.4%, respectively [68]. Receptor activator of nuclear factor kappa-B ligand (RANKL) is a cytokine instructing the differentiation of osteoclast progenitors into mature osteoclasts through various downstream cascades. In patients with bone metastasis, RANKL in bones can attract RANK-expressing circulating tumour cells to attach onto the bone matrix. In addition, this cytokine also promotes vascular permeability, which eases the escape of tumour cells into the circulation [69]. In this regard, the now commonly used targeted therapy drug denosumab is a monoclonal antibody designed against RANKL to suppress osteoclast function and preserve bone mass in patients with breast and prostate cancers [70]. Currently, there are no studies investigating the association between these biomarkers and CTCs in the prediction of bone metastases; this is a substantial knowledge gap in the state of the current research.

### 1.5. Use of Artificial Intelligence to Identify CTCs

Artificial intelligence (AI) is currently being developed to be used alongside information obtained from the analysis of CTCs. A recent development of a convolutional neural network (CNN)-based AI, to detect CTCs in patients with esophageal squamous cell carcinoma, has shown that AI’s recognition capabilities are far more superior than humans, with a reported accuracy of >99.6%. In addition, the time to count and classify CTC was also shortened by 850 times [71]. Coupled with the ability of the AI to accurately identify CTCs based on features beyond set parameters, AI can deepen our understanding of the heterogenicity and subtypes of CTCs. There are, however, limitations to using AI in the classification of CTCs, one of which is that the images utilized in AI must be fluorescently labelled. This requirement is not compatible with many CTC isolation and identification methods and so the resulting slides cannot usually be used in AI training. In addition to this, training an AI algorithm does require many images, and patient samples usually do not contain many CTCs. Even if CTCs are present in the sample, morphology and ligands can be heterogeneous, adding to the difficulties in training an AI model.

However, despite these difficulties, researchers have continued to make headway in the search for better methods to automate CTC identification. Machine learning, a subset of AI which focuses on its learning aspect, has also shown promise [72,73,74]. In one study, researchers utilized this technique to develop an ovarian cancer-specific predictive framework from peripheral blood parameters and ascertained the predictive capabilities of eight machine learning classifiers [75,76]^.^

It is clear that the use of AI in metastasis prediction promises great potential, and large-scale studies are warranted to fully validate its application in clinical settings.

## 2. Conclusions

This review summarises the current application of CTCs in the diagnosis and prediction of future bone metastasis in different cancer types, with a focus on the technologies for CTC detection, the utilization of CTCs as prognostic biomarkers and the potential of therapeutic targets. The CellSearch^TM^ system is the most commonly used technology for diagnostic and prognosis of bone metastasis in breast and prostate cancer, with the ISET system being increasingly utilised due to its lower costs, ease of handling and high sensitivity and specificity for CTC detection. The implication of CTCs in diagnosis and prognosis is now widely recognised, specifically utilising CTCs as a prognostic tool to inform risk of recurrence and therapeutic targets in bone metastasis. This review has also identified a plethora of potentially useful biomarkers which should be incorporated comprehensively into a ‘test panel’ for the prediction of bone metastasis.

## Figures and Tables

**Figure 1 cancers-16-00252-f001:**
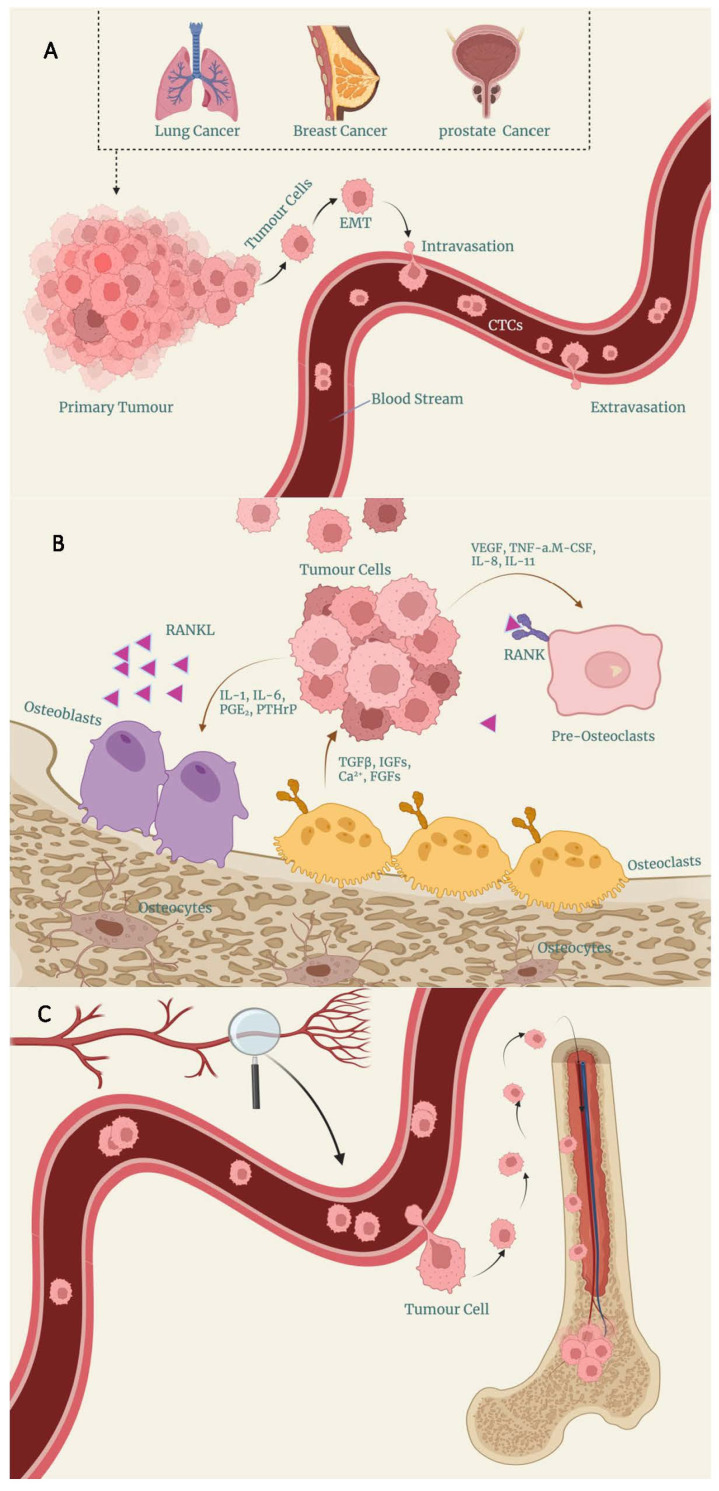
Bone metastasis of circulating tumour cells is a sequence of gradual processes that include: (**A**) Tumour cells escape from the primary tumour, undergo an epithelial-mesenchymal transition (EMT), conferring enhanced motility and invasiveness to the cells, and intravasate into the basement membrane and surrounding tissues, entering the blood circulation. (**B**) CTCs leave the vasculature by extravasation and interact with osteoclasts and osteoblasts in the bone microenvironment, leading to a local increase in tumour-derived factors that promote osteoclastogenesis and osteoblastogenesis and induce RANKL-RANK signalling in osteolytic bone metastases, thereby promoting tumour cell survival and proliferation. (**C**) Ultimately, the tumour cells complete invasion and migration, colonization and adaptation, and expanded growth.

**Table 1 cancers-16-00252-t001:** Latest CTC detection technologies in patients diagnosed with bone metastasis.

CTC Screening Technique	Enrichment Method	Blood Volume (mL)	Clinical Value	Primary Tumour Type (Study Year)
CellSearch^TM^ System	Immunomagnetic	7.5	≥5 CTCs in 90% of patients with bone metastases	Breast (2010) [30]
			Total CTCs associated with Time-to-Bone-Metastasis-Progression	Breast (2021) [31]
			≥1 CTCs in 93.65% of patients with bone metastasis	Lung (2014) [32]
			Significant association between bone metastases and CTC expressing CXCR4	Neuroendocrine (2019) [33]
			CTC number ≥2 significantly associated with liver or bone metastases	Lung (2011) [34]
			Diagnosis and prognosis of bone metastasis	Prostate (2007) [35]
CTC iChip	Immunomagnetic	10	Role of androgen receptor signalling in CTCs in breast cancer bone metastasis	Breast (2018) [36]
RT-PCR	Density gradient centrifugation	2.5	CTCs closely associated with clinical evidence of bone metastases	Prostate (2009) [37]

**Table 2 cancers-16-00252-t002:** Clinical research relating to CTC detection.

Author, Year	Primary Cancer Site	Patient Recruitment Period	No. of Patients	Technology	Study Details
De Giorgi, Valero et al., 2010 [30]	Breast cancer	2004–2008	195	CellSearch^TM^	The relationship between CTCs level and bone metastases
Basso, Facchinetti et al., 2021 [31]	Renal cancer	2008–2010	195	CellSearch^TM^	Drug trial with sunitinib (77.5%) or pazopanib (21%)
Cheng, Liu et al., 2014 [32]	Lung cancer	2009–2013	97	CellSearch^TM^	Relationship between increased CTCs and MRI-detected metastasis in bone in patients diagnosed with progressive lung cancer.
Rizzo, Vesely et al., 2019 [33]	Neuroendocrine cancer	2009–2017	254	CellSearch^TM^	Expression of CXCR4 on CTCs as a potential predictor of skeletal invasion.
Matthew G, Krebs et al., 2011 [34]	Lung cancer	2007–2009	109	CellSearch^TM^	Determine prevalence and clinical significance of CTCs in patients with non–small-cell lung cancer.
Shaffer, David R et al., 2007 [35]	Prostate cancer	2009	63	CellSearch^TM^	CTCs from peripheral blood of patients with advanced prostate cancer using immunomagnetic trapping techniques.
Aceto, Bardia et al., 2018 [36]	Breast cancer	2018	32	CTC-iChip	RNA sequencing of CTCs from with metastatic oestrogen receptor (ER)+ breast cancer, comparing cases with progression in bone versus visceral organs.
Helo, Cronin et al., 2009 [37]	Prostate cancer	2009	180	RT-PCR	CTCs in patients with localised prostate cancer or CRPC by real-time RT-PCR of KLK3 and KLK2 mRNAs.
Foroni, Milan et al., 2014 [40]	Breast cancer	2014	33	CellSearch^TM^	Anti-tumour effect of zoledronic acid (ZA)
Smid, Wang et al., 2006 [41]	Breast cancer	2016	107	-	Genes associated with breast cancer metastatic to bone
Josefsson, Larsson et al., 2018 [42]	Prostate cancer	2013–2016	25	Capturing CTCs on EpCAM- and HER2 antibody-conjugated magnetic beads	Comparing expression profiles of 41 prostate cancer-related genes between paired CTC and spinal column metastasis patients. Gene expression (EpCAM, GAPDH, GUSB, CD45, CD44)
Pantano, Rossi et al., 2020 [43]	Breast cancer	2012–2015	42	CellSearch^TM^	Novel CTC assay by using an anti-RANK monoclonal antibody in conjunction with CellSearch^TM^ platform
Onken, Fekonja et al., 2019 [44]	Lung cancer, prostate cancer, breast cancer, kidney cancer, lower gastrointestinal tract, and malignant melanoma	2005–2015	507	-	Investigating metastatic dissemination of tumour cells to spinal bone and other osseous organs
Pierga, Hajage et al., 2012 [45]	Breast cancer	2007–2009	267	CellSearch^TM^	First-line chemotherapy
De Giorgi, Mego et al., 2010 [46]	Breast cancer	2004–2008	55	CellSearch^TM^	Determining the predictive significance of CTC counts and 18F-FDG PET/CT findings in patients with bone metastases from breast cancer treated with standard systemic therapies
Lovero, D’Oronzo et al., 2022 [47]	Breast cancer	2022	30	-	Developing a targeted RNAseq assay to screen a genes critically involved in the metastatic cascade
Bidard, Vincent-Salomon, A et al., 2008 [48]	Breast cancer	1998–2005	837	-	Study of clinical outcomes in metastatic breast cancer according CTC status.
Ried, Tamanna et al., 2020 [49]	Renal cancer	2014–2019	49	ISET^®^-CTC	The cytology-based ISET^®^-CTC Test
Miyamoto, Lee et al., 2018 [50]	Prostate cancer	2018	61	CTC-iChip	A novel assay for detection of liver-derived CTCs
Mark, Rack et al., 2013 [51]	Prostate cancer	2008–2010	90	CellSearch^TM^	Explore CTC counts in different stages of prostate cancer in association with tumour burden
Danila, Heller et al., 2007 [52]	Prostate cancer	2007	120	CellSearch^TM^	Various hormonal and cytotoxic therapies.
Naito, Tanaka et al., 2012 [53]	Lung cancer	2009–2010	51	CellSearch^TM^	Evaluating relationship of CTCs to disease prognosis
Shimazu, Fukuda et al., 2016 [54]	Gastric cancer	2014–2015	39	CellSearch^TM^	Clarify the biological features contributing to gastric cancer with diffuse bone metastases at diagnosis.

## Data Availability

No new data were generated or analysed in this study. The data presented in this study are available in this article.

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
