# Peer review of "Circulating Tumour Cells in the Prediction of Bone Metastasis"

_cancers, 2024, doi:10.3390/cancers16020252_

Round 1

Reviewer 1 Report

Comments and Suggestions for Authors

Comments:

Abstract:

1.       Line 23, This review summarizes the advantages and disadvantages of the methods and in which clinical studies the specific methods have been applied.

Please improve this sentence.

2.       Line 24-27, Studies reviewed here have found that CTC numbers are predictive of bone metastasis in breast, prostate and lung tumours. Further work is required to incorporate information on CTCs into current staging 26 systems to guide treatment in the prevention of tumour progression into bone.

The sentences need to be improved.

3.       Section, 3.2. CTCs as prognostic biomarkers of recurrence and therapeutic targets for bone metastasis

This section is very lengthy; I would suggest to divide it into various subsection such as

CTC in drug screening, gene expression by CTCs, Immunotherapy and CTC.

4.       Also, there are several other markers available for bone metastasis, is there any correlation of those markers with CTCs, if yes, then it does make sense to include those markers here. Otherwise, authors may remove it.

5.       There is no figures in this manuscript. I would suggest to includes at least two attractive figures.

6.       Conclusion part is lengthy, It should be concise.

7.       Overall, throughout English editing with native speaker is required.           

Comments on the Quality of English Language

Extensive English editing is required throughout the manuscript.

Reviewer 2 Report

Comments and Suggestions for Authors

Dear Authors

Review: Cancers-2781421

Circulating Tumour Cells in the Prediction of Bone Metastasis by Wai CHOI et al summarizes the technologies used to isolate and count CTCs, and the studies investigating the role of CTC numbers and CTC cell surface protein expression in the prediction of cancer metastasis to the bone.  Also, this review summarizes the advantages and disadvantages of the methods and in which clinical studies used the specific methods. The authors found that CTC numbers are predictive of bone metastasis in breast, prostate and lung tumours.  This current review needs additional data to increase visibility of the readers.

Major Comments:

1.     This review lacking mechanistic cartoons

2.     English grammar needs to be fixed

3.     No biological marker is described

4.     Poor documentation of utilization of CTCs with Artificial Intelligence

Comments on the Quality of English Language

Proof reading is required

Round 2

Reviewer 1 Report

Comments and Suggestions for Authors

Authors addressed my concerns. therefore, I recommend it to be accepted.      

Comments on the Quality of English Language

Minor english editing is still required.